# OpenReview forum: "Unintentional Unalignment: Likelihood Displacement in Direct Preference Optimization"
_ICLR.cc/2025/Conference — ICLR 2025 Poster_

### Official Review · Reviewer_wixY · 2024-10-24

**Soundness:** 3
**Presentation:** 4
**Contribution:** 2
**Rating:** 6
**Confidence:** 3

**Summary:**

This paper studies the problem of *unintentional unalignment* in RLHF: when aligning a model with RLHF with DPO, the opposite of the desired effect can happen. For example, when aligning Llama-3-8B, we observe a drastic reduction of the refusal rate on SORRY-Bench.

This paper suggests that this phenomenon is due to embedding similarities of undesirable and desirable properties fed to the model. As a consequence, the model can only decrease the probability of both outputs, leading to the increase probability of an alternative, potentially harfmul, output.

To mitigate this effect, the authors propose a CHES score, which allows to select only a fraction of the data when running DPO. This data selection then allows to better performance on benchmarks such as SORRY-Bench.

**Strengths:**

The study of RLHF is of course an important topic, as it is implemented to align LLMs, while still presenting clear flaws, such as the one of unintentional unalignement. This work proposes a nice and simple explanation for this phenomenon, which can then lead to the design of strategies that mitigate such effect.

This work presents extended experiments on the SORRY-Bench dataset, where their curated data method allows to reach strong refusal rates.

**Weaknesses:**

While the presentation seems to insist on the theoretical contribution of this work, I find it pretty weak. More precisely, Theorems 1 and 3 just seem to be the result of the computation of a gradient. Moreover, Theorem 3 is a bit disappointing, as it depends on some factors $\alpha_{k,k'}$ for which we don't know the sign (besides an empirical study).

As a consequence, I do not see the theoretical contribution of this work as a strong contribution, and I thus expect it to be strong on the empirical side. On the empirical side now, I find the experiments on the SORRY-Bench benchmark interesting and promising. Yet, I have the following main critique regarding this work:

**RLHF has not the sole objective of maximizing the refusal rate, but at aligning with human preferences in many different aspects**

I indeed feel that this work considers the performance of a RLHF method solely on its refusal rate. Indeed, the only experiments are done wrt this metric, but I would also like to see if the proposed algorithm does not lead to poorer performances on different benchmarks (with different objective than refusal rates), such as evol-instruct.

Indeed, if the only objective was to maximize the refusal rate, then a better, obvious algorithm should be the following one: we only consider two possible values for $y^+$ and $y^-$, given by "Acceptance" and "Refusal" and fine-tune the model with such a dataset. Presumably, I guess there is no better algorithm to optimize the "refusal rate" metric, as it is explicitly and directly optimizing this metric.
On the other hand, the CHES based method does not do that explicitly, but I suspect it is close to that in the end. So why should we not directly optimize on the refusal rate? I guess such a method could lead to poorer performances on more complex alignment tasks: as a consequence, I feel that measuring the performance of the algorithm on such tasks is also needed to validate the interest of the proposed algorithm.

**Questions:**

- What are the performances of the proposed algorithm on alignment benchmarks that are not about refusal rate?
- Why could we not simply directly optimize the refusal rate?

---

> ### Author Response · Authors · 2024-11-18
> **Response (Part 1/2)**
>
> Thank you for the feedback! It seems that your critique and questions focus on the significance of our theoretical analysis and the nature of the experiments on the SORRY-Bench dataset (Section 6). We address these points below.
>
> ### **Significance of Our Theoretical Analysis**
>
> We agree that the theory is rather simple in terms of the complexity of calculations. That said, we believe that the utility of a theory need not depend on its complexity, rather on the insights and actionable guidelines that it provides.
>
> As demonstrated by the experiments of Section 3, 5, and 6, our theory identifies salient mechanisms that drive likelihood displacement — a widely observed phenomenon, which was not well understood — for contemporary models and standard preference datasets. Specifically, it characterizes how the model’s embedding geometry drives likelihood displacement, shedding light on why it can occur even in simple settings. This stands in stark contrast to prior work, which attributed likelihood displacement to various complexities in the preference learning pipeline (as discussed in Section 3, and elaborated upon in Appendix B). Furthermore, as shown in Section 5, the CHES score brought forth by our analysis accurately identifies which samples lead to likelihood displacement, whereas alternative similarity measures fail to do so (e.g., the edit distance between preferences, which was suggested by [1] as a potential source of likelihood displacement). We therefore view the relative simplicity of the analysis as a strength rather than a weakness.
>
> ### **SORRY-Bench Experiments (Section 6)**
>
> We would like to first emphasize that **the main focus of our work is on understanding why likelihood displacement occurs and its undesirable implications, as opposed to proposing a new algorithm** for improving the alignment of language models. As specified above, through a combination of theory and experiments, our results substantially advance the existing understanding of likelihood displacement and give a quantitative tool for identifying which samples are expected to cause it in a given dataset. In particular, it is not true that we include only experiments considering the refusal rate of models, as suggested by the review.
>
> The purpose of the experiments in Section 6 is to demonstrate: (i) in a clean, yet realistic setting, undesirable outcomes due to likelihood displacement; and (ii) the potential of the CHES score for mitigating such undesirable outcomes through data filtering. **By no means do we claim that one should only consider the refusal rate of a model during alignment**. Rather, the experiments demonstrate that, even when only attempting to train a model to refuse unsafe prompts, likelihood displacement can cause the refusal rates to drop (instead of increasing). As discussed in Section 6.2, this is particularly striking since there is no incentive for the model to not refuse a prompt.
>
> Note that, if one would additionally include general instruction following data, then that would only further incentivize a drop in refusal rates. To verify, we ran experiments analogous to those of Section 6 while including samples from UltraFeedback in the training set (in addition to the SORRY-Bench preference data). As expected, this led to a more severe decrease in refusal rates — e.g., the refusal rate of Llama-3-8B-Instruct dropped from 74.4% to 16.7% (in contrast to a drop to 33.4% when training only preference data from SORRY-Bench).
>
> ### **Further Exploring the Utility of the CHES Score in Additional Settings**
>
> While Sections 5 and 6 demonstrate the potential of the CHES score for mitigating undesirable effects of likelihood displacement, as stated in the conclusion section, we agree that further investigation is necessary to assess its utility more broadly (e.g., for improving performance in general instruction following settings, as measured by metrics other than the refusal rate). However, this is a non-trivial endeavor, which we believe falls outside the scope of the current work, and is more a promising direction for future research.
>
> We note that in general instruction following settings, it is difficult to categorize whether likelihood displacement is benign or catastrophic, as opposed to the safety alignment setting of Section 6, where refusal responses are benign and non-refusals are catastrophic. Thus, a major challenge is to detect additional settings where likelihood displacement is catastrophic and requires mitigation. For example, it is possible that likelihood displacement will lead to degradation in performance only for some types of prompts and evaluation metrics.
>
> ### **Conclusion**
> Thank you for taking the time to read our response. We hope that it addresses your concerns, and would greatly appreciate it if you would consider raising your score. Please let us know if you have any further questions; we will gladly expand.

---

> > ### Author Response · Authors · 2024-11-18
> > **Response (Part 2/2)**
> >
> > ### **References**
> >
> > [1] Pal, A., Karkhanis, D., Dooley, S., Roberts, M., Naidu, S., & White, C. (2024). Smaug: Fixing failure modes of preference optimisation with dpo-positive. arXiv preprint arXiv:2402.13228.

---

### Official Review · Reviewer_bfES · 2024-11-03

**Soundness:** 3
**Presentation:** 3
**Contribution:** 3
**Rating:** 8
**Confidence:** 3

**Summary:**

This work explores the causes and effects of a phenomenon in direct preference learning, referred to by the authors as likelihood displacement, where the probability of preferred responses decreases. This counterintuitive occurrence is attributed to the similarity of preference embedding, quantified using a centered hidden embedding similarity (CHES) score. This score is valuable for data selection, helping to avoid unintentional unalignment, a catastrophic phenomenon identified by the paper. The reviewer finds the paper clearly written, offering valuable insights into a commonly recognized but not well-understood issue in DPO training, with interesting and solid theoretical analysis backing the results under a simplified but reasonable mathematical setup.

**Strengths:**

1. The paper is quite clearly written, and the theoretical results for analyzing likelihood displacement are sound and well presented.
2. The paper also analyzes the variants of direct preference learning algorithms that explicitly adds SFT loss on preferred responses as regularization and shows that is can mitigate likelihood displacement.
3. The experiments demonstrate the power of the CHES score in identifying samples that lead to decreasing probability of preferred responses and its utility in data filtering to prevent unintentional unalignment.

**Weaknesses:**

1. The unconstrained feature model is not realistic and an oversimplification. But the reviewer does appreciate the attempt towards analyzing the phenomenon of likelihood displacement from a rigorous theoretical perspective.
2. The theoretical results are restricted in analyzing the local change of log probability of preferred labels, while a comprehend analysis of the entire dynamic of the changes in the log probabilities is unknown, even for the simplified single data and single output token case. Such analysis can give better characterization of whether the log preferred response probability goes up or down after direct preference learning.

**Questions:**

1. In Section 4.2.1., Line 297, why the orthogonal component of $\mathbf{W}\_{\mathbf{y}^+} - \mathbf{W}\_{\mathbf{y}^-}$ w.r.t. $\mathbf{W}\_{\mathbf{y}^+}$ is $\mathbf{W}\_{\mathbf{y}^-}$? It seems that from Theorem 1 if the likelihood displacement happens (in this specific setting of single token response), the two unembeddings should be well aligned, i.e., $\langle \mathbf{W}\_{\mathbf{y}^+} , \mathbf{W}\_{\mathbf{y}^-}  \rangle$ is relatively large, and in this case the projection to the orthogonal space of $\mathbf{W}\_{\mathbf{y}^+}$ is not likely to be introduced by $\mathbf{W}\_{\mathbf{y}^-}$. Can the authors explain more about the intuition here?
2. While the authors argue that the CHES score (Definition 2) reflects the similarity of the preferred response and the rejected response, it is a little bit vague to strictly say this from its expression because the CHES score also involves a negative term of the norm of the preferred response summed embeddings. How should one interpret such explanations made in the paper mathematically?
3. Continued from 2., from Theorem 1, the unembedding geometry also influences the change of the log preferred response probabilities. Why does the CHES score not involve terms related to the unembedding geometry of the responses?
4. Does the unintentional unalignment also appear in RLHF algorithms (other than direct preference learning, e.g., PPO)? If so, can it also be explained by the mechanism proposed by this paper?
5. What is the implication of the analyzed method in the domain of complex reasoning like math problem solving and code generation? Does the proposed data filtering method also help to improve the reasoning capability of the model trained?
6. Typos, Line 959: $\mathbf{W}\_{\mathbf{y}_!}$ should be $\mathbf{W}\_{\mathbf{y}_1}$

---

> ### Author Response · Authors · 2024-11-18
> **Response (Part 1/2)**
>
> Thank you for the feedback and for highlighting the clarity of our work and the value of our theoretical and empirical contributions! We treat your comments and questions below.
>
> > 1. The unconstrained feature model is not realistic and an oversimplification. But the reviewer does appreciate the attempt towards analyzing the phenomenon of likelihood displacement from a rigorous theoretical perspective.
> > 2. The theoretical results are restricted in analyzing the local change of log probability of preferred labels, while a comprehend analysis of the entire dynamic of the changes in the log probabilities is unknown, even for the simplified single data and single output token case. Such analysis can give better characterization of whether the log preferred response probability goes up or down after direct preference learning.
>
> We agree that the unconstrained features model simplifies the training dynamics, abstracting away the role of the architecture. However, we believe that the utility of a theory need not depend on its complexity, rather on the insights and actionable guidelines that it provides. As demonstrated by the experiments of Section 3, 5, and 6, our theory identifies salient mechanisms that drive likelihood displacement. In particular, the CHES score brought forth by it accurately identifies samples leading to likelihood displacement (Section 5), whereas alternative similarity measures fail to do so. This is a clear indication that our analysis, despite the adoption of the unconstrained features model and its local nature, captures the causes for likelihood displacement in practical settings with standard preference datasets and contemporary models. We therefore view the relative simplicity of the analysis as a strength rather than a weakness.
>
> With that said, as stated in the conclusion section, extending our theory by studying the evolution of log probabilities throughout training and how the architecture choice influences likelihood displacement (i.e. going beyond the unconstrained features model) can be fruitful directions for future work.
>
> ### **Answers to Questions**
>
> > 1. In Section 4.2.1., Line 297, why the orthogonal component of $\mathbf{W_{y^{+}} - W_{y^{-}}}$ w.r.t. $\mathbf{W_{y^{+}}}$ is $\mathbf{W_{y^{-}}}$? It seems that from Theorem 1 if the likelihood displacement happens (in this specific setting of single token response), the two unembeddings should be well aligned, i.e., $\langle \mathbf{W_{y^{+}}} , \mathbf{W_{y^{-}}} \rangle$ is relatively large, and in this case the projection to the orthogonal space of $\mathbf{W_{y^{+}}}$ is not likely to be introduced by $\mathbf{W_{y^{-}}}$. Can the authors explain more about the intuition here?
>
> This is a great question! We will gladly explain. Denoting by $\Pi_\perp$ the projection operator onto the subspace orthogonal to $\mathbf{W_{y^{+}}}$, it holds that $\Pi_\perp (\mathbf{W_{y^{+}}} - \mathbf{W_{y^{-}}} ) = - \Pi_\perp ( \mathbf{W_{y^{-}}} )$. This is what we meant by saying that $\mathbf{W_{y^{-}}}$ introduces the orthogonal component in line 296. We clarified this point in the updated manuscript; thank you for the question!
>
> Regarding when likelihood displacement occurs in the single token case, as shown in Theorem 1 and discussed in the succeeding text, there are two factors that govern when the preferred response decreases. Indeed, one of them is $\langle \mathbf{W_{y^{+}}} , \mathbf{W_{y^{-}}}\rangle$. Note that it is possible for this inner product to be relatively large and for $\mathbf{W_{y^{-}}}$ to also have a non-negligible component orthogonal to $\mathbf{W_{y^{+}}}$. As reported in Table 13, this is the case for the models and tokens considered in Section 3.

---

> > ### Author Response · Authors · 2024-11-18
> > **Response (Part 2/2)**
> >
> > > 2. While the authors argue that the CHES score (Definition 2) reflects the similarity of the preferred response and the rejected response, it is a little bit vague to strictly say this from its expression because the CHES score also involves a negative term of the norm of the preferred response summed embeddings. How should one interpret such explanations made in the paper mathematically?
> >
> > The Centered Hidden Embedding Similarity (CHES) score consists of two terms. The first is the inner product between the sum of hidden embeddings of the preferred response $\mathbf{y}^+$ and the dispreferred response $\mathbf{y}^-$, which intuitively reflects their similarity analogously to how the inner product of two vectors measures (in an unnormalized manner) their similarity. Second, is a term that “centers” this measure — the inner product of $\mathbf{y}^+$’s hidden embeddings sum with itself (i.e. its squared norm) is subtracted from the first term. The CHES score, and this centering term in particular, originates from our theoretical analysis (see Theorem 3). It accounts for the possibility that the inner product between the hidden embeddings of $\mathbf{y}^+$ and $\mathbf{y}^-$ is large just due to the hidden embeddings of $\mathbf{y}^+$ having a large norm. In that case, likelihood displacement will usually not occur since the large norm of the embeddings of $\mathbf{y}^+$, relative to their inner product with the embeddings of $\mathbf{y}^-$, will result in a larger upwards push on the log probability of $\mathbf{y}^+$.
> >
> > > 3. Continued from 2., from Theorem 1, the unembedding geometry also influences the change of the log preferred response probabilities. Why does the CHES score not involve terms related to the unembedding geometry of the responses?
> >
> > Good question! Indeed, in the multiple tokens case, the token unembedding geometry also influences whether likelihood displacement occurs. We focused on the contributions of the hidden embeddings when deriving the CHES score since, as demonstrated empirically in Sections 5 and 6, they are remarkably indicative of likelihood displacement. Thus, we did not find it necessary to derive a more complex measure for our purposes. In the future, it could be interesting to further explore the interplay between the role of the token unembedding and hidden embedding geometries.
> >
> > > 4. Does the unintentional unalignment also appear in RLHF algorithms (other than direct preference learning, e.g., PPO)? If so, can it also be explained by the mechanism proposed by this paper?
> >
> > To the best of our knowledge, likelihood displacement has only been observed for direct preference learning methods such as DPO, i.e., for objectives that depend on the gap between the log probabilities of a preferred and dispreferred response.
> >
> > > 5. What is the implication of the analyzed method in the domain of complex reasoning like math problem solving and code generation? Does the proposed data filtering method also help to improve the reasoning capability of the model trained?
> >
> > To demonstrate the potential utility of the CHES score, we considered a safety alignment scenario due to the significant harmful consequences of likelihood displacement therein. Prior work observed that DPO can also degrade the performance on math and reasoning benchmarks (e.g. [1,2]). Thus, exploring whether filtering data via the CHES score helps mitigate these degradations is indeed a valuable direction going forward. Yet, we believe that it falls outside the scope of the current paper, whose main focus is to shed light on why likelihood displacement occurs and its implications.
> >
> > > 6. Typos, Line 959: $\mathbf{W_{y_!}}$ should be $\mathbf{W_{y_1}}$
> >
> > Thank you for catching this typo! We fixed it in the updated manuscript.
> >
> > ### **Conclusion**
> >
> > Thank you for taking the time to read our response. If it addresses your concerns, we would greatly appreciate it if you would consider raising your score. Please let us know if you have any further questions; we will gladly expand.
> >
> > ### **References**
> >
> > [1] Pal, A., Karkhanis, D., Dooley, S., Roberts, M., Naidu, S., & White, C. (2024). Smaug: Fixing failure modes of preference optimisation with dpo-positive. arXiv preprint arXiv:2402.13228.
> >
> > [2] Pang, R. Y., Yuan, W., Cho, K., He, H., Sukhbaatar, S., & Weston, J. (2024). Iterative reasoning preference optimization.NeurIPS 2024.

---

> > > ### Comment · Reviewer_bfES · 2024-11-27
> > >
> > > Thank you very much for your detailed response. My concerns and questions are mostly addressed and I am glad to raise my score to 8.

---

### Official Review · Reviewer_ypEP · 2024-11-03

**Soundness:** 4
**Presentation:** 4
**Contribution:** 3
**Rating:** 6
**Confidence:** 3

**Summary:**

This paper investigates a phenomenon of “likelihood displacement” in DPO, i.e. the probability of preferred responses decreases instead of increasing, sometimes causing models to produce semantically opposite responses unintentionally. Based on the analysis, the authors propose a CHES score to identify training samples likely to cause likelihood displacement. The authors also empirically show that filtering out these samples effectively mitigates the unalignment.

**Strengths:**

1.	The paper theoretically develops a theoretical understanding of how and why likelihood displacement occurs and validates this with experiments across multiple models.

2.	CHES score provides a useful guidance to pick useful data to further enhance model performance, and potentially serves as a tool to curate preference dataset.

**Weaknesses:**

1.	The scope of the experimental scenarios is a concern. While the experiments include several models and datasets, the scenarios are somewhat limited. They primarily focus on refusal responses, which are single-token. To verify the robustness of the findings, other scenarios, including multi-token responses and diverse types of responses, are encouraged to test out.

2.	From the experiment result in Section 6, it looks like using CHES for data filtering improves the performance a lot on top of DPO. It would be great if the authors can further explore the effect of CHES filtering on more realistic tasks like finetuning on ultrafeedback dataset with models tested in the paper, compared with other baselines.

**Questions:**

1.	How does likelihood displacement evolve over longer training periods? Does it diminish or intensify as the training goes?

2.	Is there any recommended threshold for CHES filtering?

---

> ### Author Response · Authors · 2024-11-18
> **Response (Part 1/2)**
>
> Thank you for the feedback, and for highlighting the strength of our analysis and the potential of the CHES score as a preference data curation tool! We treat your comments and questions below.
>
> > 1. The scope of the experimental scenarios is a concern. While the experiments include several models and datasets, the scenarios are somewhat limited. They primarily focus on refusal responses, which are single-token. To verify the robustness of the findings, other scenarios, including multi-token responses and diverse types of responses, are encouraged to test out.
>
> We would like to point out that actually, the majority of our experiments consider settings with multi-token responses, including standard preference datasets that contain diverse response types. In particular, in Section 5 we demonstrate over the UltraFeedback and AlpacaFarm datasets that the CHES score can accurately identify samples causing likelihood displacement, whereas alternative similarity measures fail to do so. Moreover, in the experiments of Section 6, the (refusal and non-refusal) responses are not just single tokens, but rather generations either from the language model being trained (e.g. Llama-3-8B-Instruct) or from a diverse set of models.
>
> Only the experiments of Section 3 consider single-token responses. Their purpose is to demonstrate that, even in such simple settings, likelihood displacement can occur and be catastrophic, in contrast to prior work attributing likelihood displacement to various complexities in the preference learning pipeline.
>
> > 2. From the experiment result in Section 6, it looks like using CHES for data filtering improves the performance a lot on top of DPO. It would be great if the authors can further explore the effect of CHES filtering on more realistic tasks like finetuning on ultrafeedback dataset with models tested in the paper, compared with other baselines.
>
> As discussed in the conclusion section, we agree that further investigation is necessary to assess the utility of the CHES score more broadly (e.g., for improving performance in general instruction following settings via data filtering or for selecting distinct preferences from a pool of candidate responses). However, this is a non-trivial endeavor. Since the main focus of our work is on understanding why likelihood displacement occurs and its undesirable implications, we believe that further exploring uses of the CHES score falls outside the scope of the current work, and is more a promising direction for future research.
>
> We note that, in general instruction following settings, it is difficult to categorize whether likelihood displacement is benign or catastrophic, as opposed to the safety alignment setting of Section 6, where refusal responses are benign and non-refusals are catastrophic. Thus, a major challenge is to detect additional settings where likelihood displacement is catastrophic and requires mitigation. For example, it is possible that likelihood displacement will lead to degradation in performance only for some types of prompts and evaluation metrics.
>
>
> ### **Answers to Questions**
>
> > 1. How does likelihood displacement evolve over longer training periods? Does it diminish or intensify as the training goes?
>
> Empirically, we observe that when the preferred response probability starts decreasing (in some cases it can increase in the initial steps), it usually keeps decreasing for the remainder of training. This falls in line with experiments of prior work (see, e.g., Figure 5 in [1], Figures 16 and 17 in [2], and Figure 3 in [3]).
>
> > 2. Is there any recommended threshold for CHES filtering?
>
> For data filtering methods (e.g., [4,5]), the threshold that one should use usually varies depending on the setting and is selected heuristically, e.g., by considering multiple thresholds and selecting based on performance over a validation set. Similarly, we carried out experiments while filtering out different percentages of data. As noted in Section 6, keeping up to 15% of the original samples led to analogous results (i.e. an improvement in refusal rates). Beyond that, as when training on the full dataset, likelihood displacement caused refusal rates to drop. Going forward, exploring guidelines for automatically selecting such a threshold for the CHES score, and data filtering methods in general, is a valuable direction for future work.
>
> ### **Conclusion**
>
> Thank you for taking the time to read our response. If it addresses your concerns, we would greatly appreciate it if you would consider raising your score. Please let us know if you have any further questions; we will gladly expand.

---

> > ### Author Response · Authors · 2024-11-18
> > **Response (Part 2/2)**
> >
> > ### **References**
> > [1] Yuan, L., Cui, G., Wang, H., Ding, N., Wang, X., Deng, J., ... & Sun, M. (2024). Advancing llm reasoning generalists with preference trees. arXiv preprint arXiv:2404.02078.
> >
> > [2] Tajwar, F., Singh, A., Sharma, A., Rafailov, R., Schneider, J., Xie, T., ... & Kumar, A. (2024). Preference fine-tuning of llms should leverage suboptimal, on-policy data. ICML 2024.
> >
> > [3] Pang, R. Y., Yuan, W., Cho, K., He, H., Sukhbaatar, S., & Weston, J. (2024). Iterative reasoning preference optimization. NeurIPS 2024.
> >
> > [4] Xia, M., Malladi, S., Gururangan, S., Arora, S., & Chen, D. (2024). Less: Selecting influential data for targeted instruction tuning. ICML 2024.
> >
> > [5] Engstrom, L., Feldmann, A., & Madry, A. (2024). Dsdm: Model-aware dataset selection with datamodels. ICML 2024.

---

> > > ### Comment · Reviewer_ypEP · 2024-11-27
> > >
> > > Thank you for the detailed response. I understand that the primary goal of this work is to understand the phenomenon of likelihood displacement. However, I remain concerned about the practicality of the CHES score outside scenarios where the response types are limited to refusal and non-refusal. From a practical perspective, I am also curious whether applying CHES for data filtering could lead to performance improvements in DPO in a general fine-tuning task.

---

### Official Review · Reviewer_Gv88 · 2024-11-04

**Soundness:** 3
**Presentation:** 3
**Contribution:** 3
**Rating:** 8
**Confidence:** 3

**Summary:**

The paper discusses a curious and interesting phenomenon - likelihood displacement in LLMs where alignment methods actually can lead the model to be unaligned by placing much higher probability on semantically different outputs (thereby reducing the probability from preferred responses). The authors identify that likelihood displacement can be quite catastrophic, and attribute this to the geometry of the token embeddings. They introduce a metric, the Centered Hidden Embedding Similarity (CHES) score, to identify training samples that contribute most to likelihood displacement and show that filtering out high-CHES samples mitigates unalignment issues effectively.

**Strengths:**

The authors have provided quite valuable insights into the phenomenon of likelihood displacement. They have provided detailed theoretical analysis to justify their claims. I found the connection with the difference of the token unembeddings (corresponding to preferred and non-preferred outputs) quite interesting, and although it's not too hard to derive, it's good that someone pointed this out. I do this this paper will  generate quite a bit of discussion and follow up work.

**Weaknesses:**

I don't think there is any major weakness.

**Questions:**

N/A

---

> ### Author Response · Authors · 2024-11-18
> **Response**
>
> Thank you for the support and for highlighting the potential impact of our work! If any questions arise during the discussion period, please let us know and we will gladly respond.

---

### Author Response · Authors · 2024-11-24

Dear reviewers,

Thank you again for your effort and thoughtful feedback! Since the discussion period closes in a couple of days, we kindly remind you to let us know whether we have fully addressed your comments and questions. If there are any further points that you believe require clarification, we will gladly elaborate.

Best,

Authors

---

### Meta-Review · Area_Chair_5VVX · 2024-12-16

**Metareview:**

The paper points out a curious phenomenon in direct preference optimization (DPO) where the likelihood of preferred responses decreases during learning. The authors characterize this as driven by preferences that induce similar embeddings and propose the use of the centered  hidden embedding similarity (CHES) score for this. They also show empirically that filtering out training samples identified by CHES as contributing to likelihood displacement the most can mitigate this unintentional alignment. 4 out 4 reviewers recommend acceptance and I agree.

**Additional Comments On Reviewer Discussion:**

4 out 4 reviewers recommend acceptance and I agree.

---

### Decision · Program_Chairs · 2025-01-22

Accept (Poster)